# VERIFYTHISBENCH: GENERATING CODE, SPECIFICATIONS, AND PROOFS ALL AT ONCE

## ABSTRACT

Large language models (LLMs) have demonstrated remarkable progress in code generation, but many existing benchmarks are approaching saturation and offer little guarantee on the trustworthiness of the generated programs. To improve visibility into model reasoning on formal correctness, we introduce `VerifyThisBench`, a new benchmark that evaluates end-to-end program generation and proof verification from natural language descriptions: models must (i) extract formal specifications, (ii) implement in a verification-aware language, and (iii) construct machine-checkable proofs. Our evaluation reveals that even state-of-the-art (SOTA) models, such as o3-mini, achieve a pass rate of less than 4%, with many outputs failing to compile. To isolate sources of difficulty, we further propose `VerifyThisBenchXS`, a relaxed variant in which partial implementations or proofs are provided. Across nine models and seven verification tools on both benchmarks, we observe consistent gains with feedback-driven refinement, but overall pass rates remain low, underscoring substantial gaps in formal reasoning. We release the benchmark and the unified evaluation environment to catalyze the verification capabilities for future models.

## 1 INTRODUCTION

Large language models (LLMs) have unequivocally revolutionized the landscape of automated code generation. Models like OpenAI (2024) GPT-4o, Google (2024) Gemini, Anthropic (2024) Claude, and GitHub (2021) Copilot excel at generating functional code snippets and translating between languages. These capabilities are now integrated into AI-powered IDEs, such as Cursor and Visual Studio, to support large-scale software development. This proficiency had led to increasing needs on established benchmarks, as early as HumanEval (Chen et al., 2021) and MBPP (Austin et al., 2021), to reflect the capability of each LLM tool. However, this rapid progress raises critical questions about the trustworthiness and reliability of the generated artifacts. Many existing benchmarks, while useful for gauging functional correctness through test suites, are approaching saturation (Kiela et al., 2021; Ghosh et al., 2025; Gu et al., 2024; Xia et al., 2024) and inherently offer limited guarantees regarding the deeper aspects of program correctness. Test cases, by their nature, can demonstrate the presence of bugs, but cannot prove their absence (Dijkstra, 1972), leaving a significant gap in assessing the formal robustness and true reasoning capabilities of these powerful models.

Reliable software must go beyond passing tests to be trustworthy, precisely follow specifications, and even self-validate. Formal verification offers the most rigorous approach to achieving these guarantees. This paradigm involves providing machine-checked mathematical proofs to show that a program adheres to its formal specification, thereby guaranteeing critical properties such as functional correctness, liveness (ensuring the program eventually does something good), and safety (ensuring the program never does something bad) (Huth & Ryan, 2004). Modern program verification infrastructures, such as Dafny (Leino, 2010), Frama-C (Kirchner et al., 2015), Verus (Lattuada et al., 2023), Isabelle/HOL (Nipkow et al., 2002), and Lean (de Moura et al., 2018), coupled with powerful automated theorem provers and SMT solvers like Z3 (de Moura & Bjørner, 2008) and CVC5 (Barbosa et al., 2022), have significantly streamlined the process of writing and checking such verified software. These tools allow developers to express complex specifications and then automatically or semi-automatically verify that the implementation meets these specifications.

Although researchers have developed multiple benchmarks to assess LLMs on formal verification subtasks (Kamath et al., 2023; Chakraborty et al., 2023; Pei et al., 2023; Endres et al., 2024), none evaluates end-to-end program verification solely from natural-language inputs. Instead, existing suites either require verifying or synthesizing small programs against a given formal specification, or focus on aiding proof completion by suggesting individual verification steps. Consequently, even though state-of-the-art LLMs have been reported to solve up to 97.8% of these benchmark tasks (Wu et al., 2024), those numbers do not reflect their true capability for end-to-end program verification.

To bridge this gap and rigorously evaluate the capabilities of LLMs in this demanding domain, we introduce `VerifyThisBench`, a novel benchmark designed to assess end-to-end program verification, as shown in Figure 1. Inspired by the annual VerifyThis Competition Series where human contestants devise implementations and accompanying formal proofs in verification-aware languages, `VerifyThisBench` tasks LLMs with interpreting natural language problem descriptions, formulating formal specifications, generating the corresponding code, and constructing machine-checkable correctness proofs – all at once, to produce compiled and verified artifacts. While recent efforts (Ye et al., 2025; Thakur et al., 2025) also benchmark LLMs on end-to-end verification tasks in Lean, our work differs by building on the long-standing VerifyThis Challenge, offering multi-framework coverage, research-grade tasks, and competition-vetted difficulty, with solution lengths up to 648 lines compared to a maximum of 225 in prior work.

Our evaluation using `VerifyThisBench` reveals that even state-of-the-art (SOTA) models, such as o3-mini, achieve a zero-shot pass rate of 3.62% on this end-to-end task, with a significant number of outputs failing even to compile, and only reach a pass rate of 9.37% after five rounds of feedback. These results underscore the profound challenge this domain presents. To dissect these challenges further and explore capabilities in a more guided setting, we also propose `VerifyThisBenchXS`, a variant where partial specification, implementation code, or proofs are provided, and the LLM is tasked to complete the missing components. In this setting, o3-mini achieves 2.24% in zero-shot attempt and 8.28% after refinement.

This paper makes the following key contributions:

- **VerifyThisBench:** We present `VerifyThisBench`, a new benchmark suite for evaluating the ability of LLMs to generate fully verified programs (code, specifications, and proofs) from natural language descriptions.

- **Relaxed VerifyThisBench:** We introduce `VerifyThisBenchXS`, a relaxed version of the `VerifyThisBench`, to assess LLM performance when provided with partial artifacts and tasked with completing them.

- **Unified Environment:** We provide a unified evaluation environment that integrates seven verification tools and an automated pipeline, enabling consistent and scalable benchmarking across diverse formal verification tasks.

- **SOTA LLM Evaluation:** We conduct a systematic evaluation of nine SOTA LLMs on both benchmarks, revealing current capabilities and significant limitations.

## 2 BACKGROUND & RELATED WORK

### 2.1 UNVERIFIED CODE SYNTHESIS BENCHMARKS

Recent benchmarks for code generation include **APPS** (Hendrycks et al., 2021), **HumanEval** (Chen et al., 2021), **MBPP** (Austin et al., 2021), **CodeContests** (Li et al., 2022), **DS-1000**(Lai et al., 2022), **SWEBench** (Jimenez et al., 2024), and **EvalPlus** (Liu et al., 2023), among others. These benchmarks present programming tasks, often sourced from online competitions or community platforms, and evaluate models based on whether generated solutions pass a set of input-output test cases. While effective in emulating daily software development, they do not involve formal verification.

In contrast, `VerifyThisBench` requires models to go beyond functional testing: they must formalize natural language intents into specifications, generate code in verification-aware languages, and produce proofs that pass a formal logic verifier. This makes `VerifyThisBench` a substantially more rigorous and comprehensive benchmark than traditional code synthesis tasks.

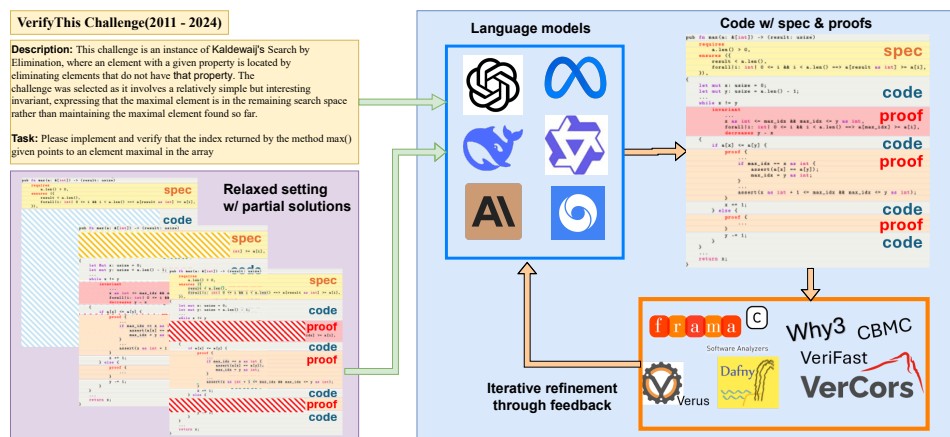

Figure 1: Evaluation workflow of `VerifyThisBench` and its relaxed settings.

## 2.2 PROGRAM VERIFICATION BENCHMARKS

Benchmarks built in the context of formal verification include **SV-COMP** (SV-COMP-org), **Sy-GuS** (Sygus-org), and **Code2Inv** (Si et al., 2020). **SV-COMP** and **Code2Inv** focus solely on verification tasks that do not require implementation generation. For more contexts, the former contains large-scale C/Java benchmarks verifying fixed safety properties, and the latter targets loop-invariant generation over small C-style programs. **SyGuS** focuses on constraint-based synthesis.

More recent efforts like **DafnyBench** (Loughridge et al., 2024) and **VerusBench** (Yang et al., 2024) collect verified programs in Dafny and Verus respectively, primarily to train and evaluate ML-based tools in aiding proof completion and suggesting verification steps, rather than end-to-end program generation from natural language.

These benchmarks evaluate components of the verification pipeline but typically assume a preset formal specification or verification goal. In contrast, `VerifyThisBench` uses the end-to-end setup to explicitly evaluate the model's ability in interpreting and encoding natural-language descriptions into provably correct formal programs, a capability not tested in existing benchmarks.

## 2.3 END-TO-END VERIFICATION BENCHMARKS

Parallel work includes **Verina** (Ye et al., 2025) (189 tasks) and **Clever** (Thakur et al., 2025) (161 tasks), exploring end-to-end verification in Lean, with sources translated from programming tasks in **HumanEval**, **MBPP** and **Leetcode** etc. `VerifyThisBench` differs in the source, scope and diversity, with 734 tasks derived from the VerifyThis competition series, which presents realistic, research-grade verification challenges across multiple domains. Rather than focusing on Lean, `VerifyThisBench` spans seven verification frameworks across multiple programming languages, including Verus (Lattuada et al., 2023) and Frama-C (Kirchner et al., 2015) that are established for production codebases. Moreover, `VerifyThisBench` includes tasks that require reasoning about memory safety, concurrency, and complex data structures beyond arrays and trees. See Appendix E for detailed comparison of description lengths and solution sizes.

## 2.4 FORMAL METHODS IN SOFTWARE VERIFICATION: A PRIMER

Formal methods in software verification aim to mathematically prove program correctness against a **formal specification**—a precise, unambiguous description of what a program should do, often expressed in a logical language. This contrasts with testing, which can only show the presence of bugs for specific inputs. The verification process typically relies on several key components embedded within or alongside the executable program code:

- Contracts: These formalize the obligations and guarantees of a code segment.

- Pre-conditions (`requires` clauses): Properties that must hold true before a function or code block executes for it to behave correctly.
- Post-conditions (`ensures` clauses): Properties guaranteed to be true after a function or code block finishes, provided its pre-conditions were met.

- Intermediate Assertions: Assistive hints are often needed to bridge any reasoning gaps between the pre&post-conditions where the underlying solver cannot automatically address.
- Loop Invariants: For iterative constructs, loop invariants are crucial properties that hold at the start of a loop, are preserved by each iteration, and, in conjunction with the loop's termination, help prove the loop's correctness.

The typical verification flow in systems utilizing these concepts is as follows:

1. **Annotation:** Developers write code in a verification-aware language (e.g., Dafny (Leino, 2010), Frama-C, Verus and annotate it with formal specifications and proof hints, including pre-conditions, post-conditions, assertions, and loop invariants.

2. **Generation of Proof Obligations:** A tool, often a Verification Condition Generator (VCG), processes the annotated code and its specifications. It translates them into a series of mathematical proof obligations (verification conditions) that, if all true, logically imply the program's correctness with respect to its specification.

3. **Automated Proving:** These verification conditions are then fed to backend automated theorem provers, typically Satisfiability Modulo Theories (SMT) solvers like Z3 (de Moura & Bjørner, 2008) or CVC5 (Barbosa et al., 2022). These solvers attempt to mathematically prove each obligation.

4. **Feedback:** The system reports to the developer whether the proofs succeeded or failed. Failures often pinpoint inconsistencies between the code and its specification, or missing/incorrect annotations.

Successfully generating code within this paradigm, as targeted by our `VerifyThisBench`, requires an LLM not only to produce the algorithmic implementation but also to understand, formulate, and correctly express intricate formal specifications and proof structures that enable automated verification.

## 3 VERIFYTHISBENCH BENCHMARK

`VerifyThisBench` is inspired by the annual **VerifyThis Challenges** (VerifyThis Competition Series), a competition where participants are tasked with formalizing specifications, implementing solutions, and verifying that the implementations meet the specification. We focus on this benchmark because it is a dedicated formal methods competition, designed not only to evaluate participants' skills but also to assess the maturity of verification tools (Denis & Siegel, 2024). Each challenge is designed to be completed within a 90-minute session and varies in difficulty. Submissions are evaluated based on correctness, completeness, and additional quality criteria such as elegance and the degree of automation. Similarly, in `VerifyThisBench`, the task is to interpret natural language problem descriptions and implement code and write proofs.

### 3.1 BENCHMARK CONSTRUCTION

We collected challenges from the annual competition series between 2011 and 2024, each with natural-language descriptions (seldom include pseudo-code) and associated (one or more) tasks. Tasks are categorized as either implementation (completing an algorithm) or verification (proving a model or implementation correct against a specification). The resulting dataset includes 41 challenges and 154 tasks, with an example in Appendix H and detailed statistics in Appendix E. The dataset is available in supplementary material and will be made public after the anonymity period.

### 3.2 ENVIRONMENT

To facilitate evaluation, we provide a unified environment supporting seven verification tools. Five of them, **Dafny** (Leino, 2010), **Why3**, **VeriFast**, **VerCors**, and **Frama-C** (Kirchner et al., 2015),

are widely used in past VerifyThis competitions. To broaden tool diversity, we additionally include **Verus** (Lattuada et al., 2023) and **CBMC** (Kroening et al., 2023), covering Rust, C, and other imperative or deductive platforms. Tool versions and brief descriptions can be found in Appendix C.

## 3.3 FEATURES OF VERIFYTHISBENCH

**End-to-end verification tasks with natural language problem descriptions**: All tasks start with informal, natural language prompts (often with pseudo-code). Models must interpret the intent and formalize it into precise logical specifications. They are required to generate specifications, implementations, and formal proofs in a verification-aware language, ensuring the code passes machine-checkable verification. Example challenge and solution can be found in Appendix H.

**Graded difficulty and multi-step challenges:** Challenges are drawn from the VerifyThis competition and span a range of difficulties (see Appendix E.). Many include sequential subtasks, allowing fine-grained assessment of model capability on step-wise tasks.

**Tool diversity:** Multiple tools are provided and tested on. Models must conform to the syntax and semantics of real-world verification frameworks.

## 3.4 RELAXATION

We observe that most language models fail to generate compilable code when targeting specific verification tools. This is often due to the syntactic complexity and precise annotations required by these tools. To isolate the sources of difficulty and better assess LLM capabilities under more supportive conditions, we construct a set of relaxed subtasks derived from past human-written solutions. Specifically, we define three forms of relaxation. In **Code-Gen**, we provide the function specifications, omitting both the implementation and the proof annotations. In **Specification-Gen**, we provide the implementation and its proof, but remove the function specifications. In **Loop-Gen**, we provide specifications and implementations, but remove loop invariants needed for verification.

In total, we create a set of 580 tasks. Specifically, there are 226 code-gen task, 121 loop-gen tasks, and 233 spec-gen tasks. Table 6 in Appendix A shows the statistics of `VerifyThisBenchXS`. Since no prior solutions exist for CBMC and Verus, and given notable community interest, we developed new Verus solutions to enrich the dataset; CBMC solutions remain unavailable and are therefore not included in the relaxed experiments.

## 4 EXPERIMENT RESULTS

### 4.1 MODEL SETUP

We evaluate a diverse set of SOTA language models, covering both proprietary and open-source systems. Representatives are selected from the OpenAI (2025) family (GPT-4o, GPT-4omini, o3-mini, o4-mini), Anthropic (2025) (Claude-3.7-Sonnet), Google (Gemini-2.5-Flash) (DeepMind, 2025), DeepSeek (Deepseek-chat-v3) (DeepSeek-AI, 2024), Meta (Llama3.3-70B-Instruct) and Alibaba (Qwen-2.5-72B-Instruct) (Qwen, 2024). This selection enables a comprehensive comparison across different model architectures and training paradigms. Model versions are provided in Appendix B.

### 4.2 EXPERIMENT DESIGN AND METRICS

For both `VerifyThisBench` and `VerifyThisBenchXS`, we conduct experiments with iterative refinement based on tool-generated error messages. To evaluate correctness, we pass the generated code to the target verification tool and check whether it compiles and verifies successfully. A task is marked as pass/succeed if no error is returned.

In addition to correctness checking, we introduce a coherence check as a relaxed evaluation metric. In this step, the model self-assesses whether its generated code semantically aligns with the original problem intent – an aspect difficult to verify automatically. This metric helps determine how well the specification matches the task description and provides insight into the model's ability in auto-formalization and symbol grounding.

Each task is attempted five times per model. The first attempt uses only the task prompt; the next four incorporate feedback from previous errors. During refinement, the model has access to the full history of its prior attempts and corresponding feedback for the current task, enabling iterative correction.

In `VerifyThisBench`, a challenge may have multi-stage tasks that are completed sequentially. Only the final attempt from the previous subtask is carried over to the next, preserving essential contexts while keeping the prompts concise. In contrast, `VerifyThisBenchXS` tasks have isolated contexts and are completed independently, with no progress carried over between tasks.

To ensure fairness, we use the same prompt (see Appendix D) across all models and set the temperature to 0.7 when applicable. Timeout of one minute is enforced for all experiments on the verifiers. The experiments were conducted on a machine with an Intel i7-1360P CPU and 16GB of RAM.

## 4.3 OVERALL PASS RATE

Table 1 presents the performance of the SOTA models on `VerifyThisBench`. For each verification tool, we report pass rates on the initial zero-shot attempt and after four additional refinement attempts using feedback.

In the first attempt, most models perform poorly, with success rates under 4%. The top performers are o3-mini, Llama, and Claude, indicating that even the strongest models struggle initially. By the fifth attempt, performance improves significantly across all models. o3-mini leads overall, followed by Claude, o4-mini, and Llama. These results highlight the effectiveness of iterative refinement and feedback in enhancing model performance.

Each model exhibits distinct strengths across different verification tools, underscoring that no single model consistently outperforms the rest. For example, o3-mini, the top overall performer, excels especially in CBMC and Verus. On the other hand, Claude shows consistent strength in Dafny and Frama-C. Gemini, while generally average, performs exceptionally well on VerCors. Llama, another open-source model, performs best on Verus. In contrast, Qwen shows consistently low performance across all tools, suggesting limitations in its current proof synthesis capabilities. Further insights into tool-specific performance are discussed in Section 4.6.

Table 1: Overall Pass Rate On `VerifyThisBench`

|  | Attempt | GPT4o | GPT4o-mini | o3-mini | o4-mini | Claude | Gemini | Llama | Deepseek | Qwen |
|---|---|---|---|---|---|---|---|---|---|---|
| CBMC | zero-shot | **8.44%** | 7.14% | **8.44%** | 1.30% | 6.49% | 1.95% | 7.14% | 0.65% | 1.30% |
|  | refinement | 20.13% | 19.48% | **25.32%** | 15.58% | 22.08% | 14.94% | 20.13% | 22.08% | 3.25% |
| Dafny | zero-shot | 0 | 0 | **4.55%** | 1.95% | 3.25% | 0 | 0 | 1.30% | 0 |
|  | refinement | 1.30% | 0.65% | 9.74% | **10.39%** | 9.74% | 1.30% | 2.60% | 2.60% | 0.65% |
| Frama-C | zero-shot | 0 | 0.65% | 0 | 0 | **3.90%** | 0.65% | 0 | 1.95% | 0 |
|  | refinement | 7.14% | 1.95% | 2.60% | 3.25% | **11.04%** | 8.44% | 0.65% | 3.25% | 0 |
| VerCors | zero-shot | 0 | 1.30% | 1.30% | 1.95% | 0 | 5.84% | **8.44%** | 1.30% | 0 |
|  | refinement | 1.95% | 1.30% | 1.95% | 5.19% | 1.30% | **16.88%** | 11.69% | 3.90% | 3.25% |
| VeriFast | zero-shot | 0 | 0 | 0 | 0 | 0 | 0 | 0 | 0 | 0 |
|  | refinement | 0 | 0 | 0 | **2.60%** | 0 | 0 | 0.65% | 0.65% | 0 |
| Verus | zero-shot | 1.95% | 6.49% | **10.39%** | 0.65% | 0.00% | 0.65% | 7.79% | 0.65% | 0.65% |
|  | refinement | 12.99% | 9.09% | **21.43%** | 8.44% | 0.65% | 0.65% | 17.53% | 1.30% | 0.65% |
| Why3 | zero-shot | 0 | 0 | 0.65% | 0.65% | **1.30%** | **1.30%** | 0 | 0.65% | 0 |
|  | refinement | 0 | 0 | 4.55% | **10.39%** | 9.09% | 5.84% | 1.95% | 1.95% | 0 |
| Overall | zero-shot | 1.48% | 2.23% | **3.62%** | 0.93% | 2.13% | 1.48% | 3.34% | 0.93% | 0.28% |
|  | refinement | 6.22% | 4.64% | **9.37%** | 7.98% | 7.70% | 6.86% | 7.88% | 5.10% | 1.11% |
| Improvement |  | 4.73% | 2.41% | 5.75% | **7.05%** | 5.57% | 5.38% | 4.55% | 4.17% | 0.83% |

Table 2 shows the results on `VerifyThisBenchXS`. Similarly, at the first attempt, absolute numbers remain low (less than 4%) for all models. At the fifth iteration, o4-mini tops the competition with 17.24%, followed closely by Deepseek (16.72%), Claude (16.03%), and Llama (11.55%). Feedback leads to substantial improvement for most models, achieving relative gains of over 10%.

In conclusion, while few models succeed from scratch, many become competitive when guided by partial context. Open-source models like Deepseek, and Llama outperform many closed-source counterparts, showing strong potential for real-world deployment in assisted formal verification. These results also underscore the importance of combining structural hints, feedback loops, and domain-specific strengths when applying LLMs to formal reasoning tasks.

Table 2: Overall Pass Rate On `VerifyThisBenchXS`

| | Attempt | GPT4o | GPT4o-mini | o3-mini | o4-mini | Claude | Gemini | Llama | Deepseek | Qwen |
|---|---|---|---|---|---|---|---|---|---|---|
| Dafny | zero-shot | 0 | 1.35% | **2.70%** | 1.35% | 1.35% | 1.35% | 0 | **2.70%** | 1.35% |
| | refinement | 17.57% | 9.46% | 31.08% | 37.84% | **41.89%** | 17.57% | 8.11% | 21.62% | 8.11% |
| Frama-C | zero-shot | 0 | 0 | 1.85% | 0 | 1.85% | 1.85% | 0 | 1.85% | 0 |
| | refinement | 0 | 0 | 5.56% | **18.52%** | 9.26% | 1.85% | 0 | 5.56% | 0 |
| VerCors | zero-shot | 0 | 0 | 0 | 0 | 0 | 0 | 0 | 0 | 0 |
| | refinement | 0 | 0 | 0 | **7.69%** | 0 | 0 | 3.85% | 0 | 0 |
| VeriFast | zero-shot | 7.58% | 4.55% | 3.03% | 6.06% | 6.06% | 0 | 4.55% | **12.12%** | 3.03% |
| | refinement | 12.12% | 6.06% | 4.55% | 10.61% | **27.27%** | 0 | 9.09% | 13.64% | 6.06% |
| Verus | zero-shot | 7.07% | 5.05% | 8.08% | 14.14% | 14.14% | 4.04% | 3.03% | 7.07% | 5.05% |
| | refinement | 15.15% | 6.06% | 17.17% | 30.30% | 30.30% | 16.16% | 13.13% | 20.20% | 7.07% |
| Why3 | zero-shot | 0 | 0 | 0 | 0.38% | 0.38% | **0.77%** | 0 | 0.38% | 0 |
| | refinement | 7.66% | 2.30% | 0.77% | 8.81% | 3.45% | 1.15% | 15.71% | **18.77%** | 1.15% |
| Overall | zero-shot | 2.07% | 1.55% | 2.24% | 3.45% | **3.62%** | 1.38% | 1.03% | 3.28% | 1.38% |
| | refinement | 9.66% | 3.97% | 8.28% | **17.24%** | 16.03% | 5.69% | 11.55% | 16.72% | 3.45% |
| Improvement | | 7.59% | 2.42% | 6.04% | **13.79%** | 12.41% | 4.31% | 10.52% | 13.44% | 2.07% |

**Key Insights**: Average pass rates for all evaluated models remain low at 10% on `VerifyThisBench` and 18% on `VerifyThisBenchXS`, revealing the challenges formal verification poses even to SOTA LLMs. All models improve with feedback.

## 4.4 FAILURE MODE DISTRIBUTION

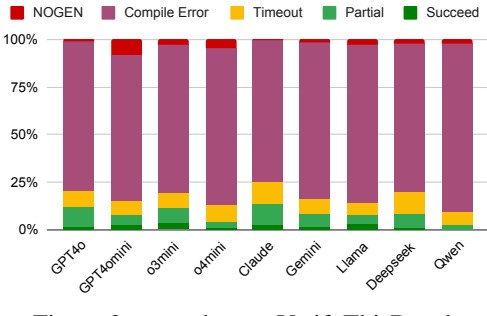

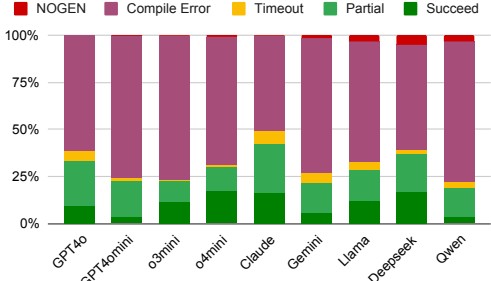

Figure 2: zero-shot on VerifyThisBench

Figure 3: refinement on VerifyThisBench

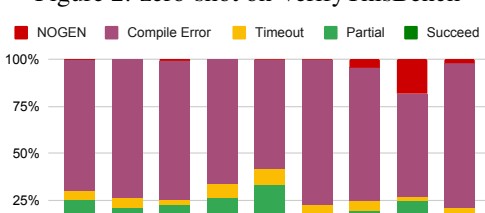

Figure 4: zero-shot on VerifyThisBenchXS

Figure 5: refinement on VerifyThisBenchXS

We categorize outcomes as **NOGEN** (no code detected), **Compile Error**, **Timeout** (compiles but exceeds verifier time budget), **Partial** (some but not all obligations proved), and **Succeed**. Figures 2 to 5 show clear improvements in model's performance when partial solution templates are provided in the relaxed settings.

Specifically, **partial** success rates increase significantly, indicating that template hints help models generate more accurate solutions. **Timeout** rates remain relatively stable. This state indicates that models are making meaningful progress toward valid proofs, but the verifier struggles to find counterexamples on difficult obligations. **Compilation errors** still dominate but tend to decrease under the relaxed setting for some models, demonstrating that not needing to generate from scratch helps reduce syntax-level mistakes. However, some models like GPT4o-mini and o3-mini exhibit

mixed trends, suggesting that while the template helps, the model's internal understanding and code generation fidelity still vary.

If we relax the metric to consider compilable code rather than fully verified solutions, Claude, GPT-4o, and Deepseek consistently emerge as the top performers across both benchmarks. Notably, Claude generates compilable outputs in nearly 50% of attempts on `VerifyThisBenchXS` and around 25% on `VerifyThisBench` in the first attempt alone, highlighting its strong baseline capability even without iterative feedback.

> **Key Insights**: While compilation error dominates in both benchmarks, in the relax setting we observe decreases in such failures and increases of partial correct or compilable solutions, moving model performance closer to usable verification outputs even when full correctness is not achieved.

## 4.5 COHERENCE

Table 3 reports each model's coherence confidence, i.e. whether the model believes its generated specification matches the intended problem requirement. Importantly, this "self" alignment assessment is computed in a separate pass without chain-of-thought disclosure of how the answer was generated by the model and is thus a statistically independent evaluation. This metric is evaluated across the verified fraction of the outputs. While passing a formal verifier indicates syntactic and logical correctness, it does not address the alignment problem (i.e., whether the verified implementation perfectly aligns with the user-intent expressed in natural language descriptions); hence, coherence offers complementary insight. Notably, except o3-mini and Qwen, models's confidence is less than 50% on passed solutions.

The results reveal considerable variance across models in their self-assessment behavior. Models like o3-mini and Claude exhibit high confidence, often reporting over 80% coherence even in the zero-shot setting, suggesting strong internal certainty—though this may reflect overconfidence rather than accurate introspection. In contrast, models like GPT-4o and Llama show much more conservative estimates, with coherence below 30%, indicating either better-calibrated uncertainty or limited self-awareness. Interestingly, refinement tends to reduce overconfidence for some models (e.g., Claude) while slightly improving coherence calibration for others (e.g., GPT-4o and Deepseek), suggesting iterative attempts help align perceived and actual correctness.

Table 3: Self-Assessment of Specification Coherence on `VerifyThisBench`

|  | Attempt | GPT4o | GPT4o-mini | o3-mini | o4-mini | Claude | Gemini | Llama | Deepseek | Qwen |
|---|---|---|---|---|---|---|---|---|---|---|
| CBMC | zero-shot | 15.38% | 0% | 84.62% | 0% | 80.00% | 33.33% | 0% | 100% | 100% |
|  | refinement | 16.13% | 0% | 61.54% | 12.50% | 26.47% | 13.64% | 3.23% | 20.59% | 100% |
| Dafny | zero-shot | - | - | 100% | 100% | 100% | - | - | 100% | - |
|  | refinement | 50.00% | 100% | 100% | 62.50% | 76.47% | 50.00% | 25.00% | 100% | 0% |
| Frama-C | zero-shot | - | 100% | - | - | 100% | 0% | - | 100% | - |
|  | refinement | 100% | 100% | 75.00% | 80.00% | 70.59% | 53.85% | 0% | 100% | - |
| VerCors | zero-shot | - | 100% | 100% | 33.33% | - | 66.67% | 69.23% | 100% | - |
|  | refinement | 66.67% | 100% | 100% | 62.50% | 0% | 46.15% | 61.11% | 85.71% | 80.00% |
| VeriFast | zero-shot | - | - | - | - | - | - | - | - | - |
|  | refinement | - | - | - | 0% | - | - | 0% | 100% | - |
| Verus | zero-shot | 0% | 30.00% | 100% | 0% | 0% | 0% | 8.33% | 0% | 0% |
|  | refinement | 0% | 28.57% | 93.94% | 7.69% | 0% | 0% | 3.70% | 0% | 0% |
| Why3 | zero-shot | - | - | 100% | 100% | 100% | 0% | - | 100% | - |
|  | refinement | - | - | 100% | 53.33% | 35.71% | 22.22% | 0% | 100% | - |
| Average | zero-shot | 12.50% | 25.00% | 94.87% | 50.00% | 88.00% | 43.75% | 27.78% | 90.91% | 66.67% |
|  | refinement | 28.36% | 20.00% | 82.00% | 36.47% | 45.35% | 34.25% | 16.47% | 46.43% | 75.00% |

We manually inspected a subset of successful solutions to validate if generated specifications align with the intended problem. Except for o3-mini, most models appear honest in their coherence self-assessments, with no false negatives found. Thus, our evaluation reflects an optimistic upper bound on true alignment—assuming coherence estimates are accurate and verifier passes indicate best-case correctness. Automatically verifying the alignment between generated specifications and user intent in natural language remains an open technical challenge (Lahiri, 2024). Our benchmark serves as a valuable resource for systematically investigating this specification–intent alignment problem in

future research. In addition, we explore a test-based evaluation approach, with preliminary results presented in Appendix F

---

**Key Insights**: Models show a wide range in coherence confidence level, suggesting varied internal behaviors. On average, only 43% of passed solutions are judged coherent and our manual review suggests strong alignment.

---

### 4.6 PERFORMANCE BY TOOLS

Table 4 shows that all tools benefit from iterative refinement through feedback. In the `VerifyThisBench` setting, CBMC and Verus exhibit the most pronounced improvements, likely due to their syntactic resemblance to C and Rust, making them more accessible to language models. Dafny also shows moderate gains in this setting. In `VerifyThisBenchXS`, improvements are even more substantial. Dafny, in particular, demonstrates a leap from near-zero success rate to over 21.4%; Verus observes an improvement around 10%. In contrast, tools such as VeriFast, Frama-C, and Why3 remain largely stagnant on both benchmarks, suggesting either stricter syntactic or semantic constraints, or a structural mismatch with current model capabilities.

Table 4: Average Pass Rates across Tools

|  | Attempt | CBMC | Dafny | Frama-C | VerCors | VeriFast | Verus | Why3 |
|---|---|---|---|---|---|---|---|---|
| `VerifyThisBench` | zero-shot | **4.76%** | 1.30% | 0.79% | 2.31% | 0 | 3.32% | 0.51% |
|  | refinement | **18.11%** | 4.47% | 4.26% | 5.34% | 0.43% | 8.15% | 3.75% |
| `VerifyThisBenchXS` | zero-shot | - | 1.35% | 0.82% | 0 | 5.22% | **7.52%** | 0.21% |
|  | refinement | - | **21.47%** | 4.53% | 1.28% | 9.93% | 17.28% | 6.64% |

### 4.7 PERFORMANCE BY RELAXATION

Table 5: Overall Performance across Different Relaxation Settings in `VerifyThisBenchXS`

| | Code | | Specification | | Loop | |
|---|---|---|---|---|---|---|
| Model | Zero-shot | Refinement | Zero-shot | Refinement | Zero-shot | Refinement |
| GPT4o | 0.88% | 11.06% | 3.00% | 9.87% | 2.48% | 6.61% |
| GPT4omini | 0.88% | 3.98% | 2.15% | 3.86% | 1.65% | 4.13% |
| o3mini | 0.88% | 7.52% | 2.58% | 7.72% | 4.13% | 10.74% |
| o4mini | 0.88% | 14.16% | **5.15%** | 18.45% | 4.96% | **20.66%** |
| Claude | **2.21%** | 15.04% | 4.29% | **19.31%** | 4.96% | 11.57% |
| Gemini | 1.33% | 6.19% | 1.29% | 4.72% | 1.65% | 6.61% |
| Llama | 0.44% | 11.95% | 1.72% | 12.88% | 0.83% | 8.26% |
| Deepseek | 0.44% | **15.49%** | 4.72% | **19.31%** | 5.79% | 14.05% |
| Qwen | 1.33% | 3.54% | 1.29% | 3.86% | 1.65% | 2.48% |
| Overall | 1.05% | 9.73% | 2.90% | **11.27%** | 3.20% | 9.81% |

Table 5 breaks down `VerifyThisBenchXS` results by **Code-Gen**, **Spec-Gen**, and **Loop-Gen**. Iterative refinement consistently improves pass rates across all categories.

Among the three, spec-gen yields the highest overall pass rates, suggesting that models can more readily articulate reasoning about what a program is supposed to do, given a working implementation and its proof context. Completing loop invariant, arguably the most abstract and logically demanding task, results in pass rate lower than 10%, though still showing solid gains with retries. This points to the inherent difficulty models face in understanding and completing partial proofs.

---

**Key Insights**: Generating the entire solution holistically (overall pass rate@9.73%) may not be more difficult than generating a specific one, e.g., loop invariant (overall pass rate@9.81%).

---

## 5 CONCLUSION

In this work, we introduce **VerifyThisBench** and **VerifyThisBenchXS** to evaluate the formal verification capabilities of large language models, systematically assessing their performance

across a range of tools, tasks, and relaxation settings. Despite the use of SOTA models, results show generally poor performance, particularly in strict end-to-end settings that require complete formal reasoning without assistance. These findings highlight significant gaps in current models' ability to generate semantically and logically correct solutions in formal domains.

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

## A COMPOSITION OF VERIFYTHISBENCHXS

Table 6 presents the composition of `VerifyThisBenchXS`, summarizing the number of verification tasks by tool and category. It includes counts of implementations, specifications, and loop-related completion tasks for six verification tools: Dafny, Frama-C, VerCors, Verifast, Why3, and Verus. In total, the benchmark comprises 580 tasks, with 226 implementations, 233 specifications, and 121 loop invariants related examples.

Table 6: Composition of `VerifyThisBenchXS`

| Tool | Implementaion | Specification | Loop | Total |
|------|---------------|---------------|------|-------|
| Dafny | 28 | 25 | 21 | 74 |
| Frama-C | 15 | 15 | 24 | 54 |
| VerCors | 8 | 8 | 10 | 26 |
| VeriFast | 26 | 31 | 9 | 66 |
| Why3 | 118 | 117 | 26 | 261 |
| Verus | 31 | 27 | 31 | 99 |
| Total | 226 | 233 | 121 | 580 |

## B MODEL VERSIONS

GPT-4o was evaluated using the version from August 6, 2024, while GPT-4o-mini and o4-mini correspond to the July 18, 2024 versions. The o3-mini model was accessed as of January 31, 2025. Claude refers to the Claude-3.7-Sonnet version released on February 24, 2025, and Gemini-2.5 Flash on the April 17, 2025 release. For open-source models, we used LLaMA3.3-70b Instruct from December 6, 2024, DeepSeek-chat-v3 from March 24, 2025, and Qwen2.5-72b Instruct from September 19, 2024. These version references ensure the reproducibility and consistency of our benchmarking results.

## C TOOL VERSIONS

We report exact toolchain versions for reproducibility and summarize each tool's verification model. The Verus verifier was run using version v0.2025.04.03.0f22710, while Why3 was evaluated with version v1.6.0. For Frama-C, we used version v30.0, and VeriFast experiments were conducted with version v25.02. The Dafny toolchain ran on version v4.10.0, and VerCors with v2.3.0. Finally, we used CBMC version v6.5.0.

Docker container images and unified toolchain launch scripts are included in the released dataset. Below we briefly describe each tool:

- **Dafny:** A verification-aware programming language with built-in specification support (pre/post-conditions, invariants) and an automatic static verifier.
- **Why3:** A platform for deductive verification with its own intermediate language (WhyML) and integration with external theorem provers.
- **VeriFast:** A verifier for C and Java using separation logic, enabling modular reasoning about memory safety and functional correctness.
- **VerCors:** A verifier for concurrent programs in Java, C, and OpenCL, supporting permission-based separation logic and parallel reasoning.
- **Frama-C:** A modular analysis platform for C, using the ACSL specification language and combining abstract interpretation with deductive verification.
- **Verus:** A verifier for Rust programs that checks user-defined specifications using SMT solving, supporting low-level features and ownership semantics.
- **CBMC:** A bounded model checker for C and C++ that verifies safety and functional correctness by translating code into SAT/SMT formulas.

## D PROMPT FORMATS

As prompt optimization was not the focus of this work, we used a simple, uniform structure for all models to ensure fairness across different tools. Each prompt consists of a system prompt describing the verification tool, followed by the problem description and task. System prompts used in our experiments are included in the released dataset (see artifact).

**(1) System prompt**: a concise tool description and key syntax/semantics reminders.

```
1 You are an assistant that writes formally verified programs in <TOOL>.
2 - Use <language/syntax> with pre/postconditions, assertions, and loop invariants as required.
3 - The solution must compile and pass the <TOOL> verifier with a 60s timeout.
4 - Do not use unsupported features: <list>.
5 - Return a single <file-type>, with all annotations needed for verification.
```

**(2) User prompt**: the natural-language problem overview and the specific task.

```
1 # Description
2 <Problem overview in natural language; may include pseudo-code.>
3
4 # Task
5 <Explicit instruction: implement/specify/prove/refine the desired property.>
```

## E STATISTICS OF VERIFYTHISBENCH AND VERIFYTHISBENCHXS

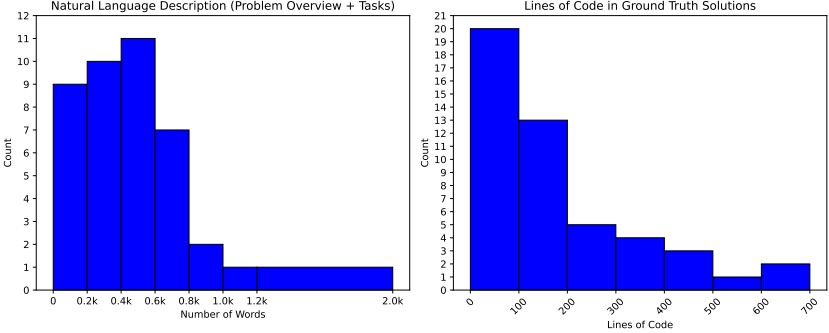

Figure 6: Distributions of dataset characteristics. (Left) Word count distribution of natural language descriptions for challenges. (Right) Line-of-code distribution of collected ground-truth solutions.

To provide empirical support for our claim regarding the range of difficulty in the dataset, we report several descriptive statistics. The natural language descriptions (problem overview and task statements) vary substantially in length, with an average of 467 words, ranging from 30 to 1802 words. The distribution, shown in Figure 6 (left), indicates that most challenges fall within the 200–799 word range, with a small number extending beyond 1000 words. In terms of solution complexity, we analyzed 48 collected ground-truth implementations, which range from as short as 28 lines to 648 lines, with an average of 189.44 lines and a median of 124 lines per solution. As illustrated in Figure 6 (right), the majority of these solutions are under 300 lines, with a few extending beyond 500 lines.

Beyond length, the diversity of task types also reflects difficulty variation: 15 out of 41 challenges involve relatively simple data structures such as binary trees and one-dimensional arrays, whereas the remaining challenges address more complex structures, including linked lists, graphs, queues, and specialized task-specific data types. Additionally, 11 out of 41 challenges explicitly require memory safety proofs, further illustrating the technical depth of the dataset.

Natural language descriptions in Verina (Ye et al., 2025) have a median length of 110 and max length of 296 words, with accompanying code and specifications of up to 100 lines. Clever (Thakur et al., 2025) reports proof lengths ranging from 10 to 225 lines. In contrast, our benchmark spans a much broader range of difficulty.

## F  EXPLORATION OF TEST-BASED SPECIFICATION VERIFICATION

Inspired by parallel work (Ye et al., 2025), we further explore a test-based proxy to evaluate specification alignment. We manually construct desired input-output pairs of a problem, and verify them against the specifications generated by the models. We check if the *inputs* imply the described *pre-conditions*, and if the outputs satisfy the post-conditions. Our setup supports open-ended, complex verification problems, without restrictions on how the function signatures or data structures are defined. As a preliminary experiment, we evaluated on all passed or failed samples of Dafny specifications generated from the `VerifyThisBench` end-to-end tasks using test cases, on the following two benchmark problems:

1. Finding the maximum in an array, and
2. Finding the maximum in a tree.

For the array version, 87.5% of the generated specifications passed the test cases (as a reference, the model's self-assessment on coherence: 93%). For the tree version, only 10% passed, mainly due to syntax errors, helper function verifiability, and other issues (reference on the model's self-assessment on coherence: 87%).

These results differ from our manual evaluations and the model's self-assessments. Model's assessment focuses on intent alignment, whereas testing requires functional correctness. This illustrates the complementary nature of different evaluation methods.

## G  VERIFYTHISBENCHXS DATA SOURCE

Table 7 lists the sources of solutions used to construct `VerifyThisBenchXS`. It includes the year of publication, the name of the verification challenge, the verification tool used, and the authors or contributors of each solution. We include canonical community solutions where available; in addition to the list, we contribute new Verus solutions (see Section 3.4).

Table 7: Solution used to generate `VerifyThisBenchXS`.

| Year | Challenge Name | Tool | Authors |
|---|---|---|---|
| 2024 | The Rope Data Structure | Why3 | Jean-Christophe Filliâtre |
| 2024 | Smart Array Copy by Shuffled Subsegments | Why3 | Jean-Christophe Filliâtre |
| 2023 | Binary Decision Diagrams | Why3 | Martin Clochard and Yannick Moy |
| 2021 | Lexicographic Permutations | Why3 | Jean-Christophe Filliâtre and Andrei Paskevich |
| 2021 | Lexicographic Permutations | VerCors | Marieke Huisman and Sebastiaan Joosten |
| 2021 | DLL to BST | Why3 | Jean-Christophe Filliâtre and Andrei Paskevich |
| 2021 | Shearsort | Why3 | Jean-Christophe Filliâtre and Andrei Paskevich |
| 2019 | Monotonic Segments and GHC sort | Frama-C | Virgile Prevosto and Virgile Robles |
| 2019 | Monotonic Segments and GHC sort | Dafny | Sample answer from report |
| 2019 | Cartesian Trees | Frama-C | Virgile Prevosto and Virgile Robles |
| 2019 | Sparse Matrix Multiplication | Frama-C | Virgile Prevosto and Virgile Robles |
| 2018 | Array Based Queuing Lock | Why3 | Raphael Rieu |
| 2018 | Gap buffer | Why3 | Raphael Rieu |
| 2018 | Colored tiles | Why3 | Raphael Rieu |
| 2017 | Pair Insertion Sort | Frama-C | Lionel Blatter and Jean-Christophe Léchenet |
| 2017 | Pair Insertion Sort | Dafny | Jon Mediero Iturrioz |
| 2017 | Pair Insertion Sort | VerCors | Marieke Huisman, Wytse Oortwijn |
| 2017 | Maximum-sum Array(one-dimension) | Frama-C | Lionel Blatter and Jean-Christophe Léchenet |
| 2017 | Odd-even Transposition Sort | Frama-C | Lionel Blatter and Jean-Christophe Léchenet |
| 2017 | Tree Buffer | Frama-C | Lionel Blatter and Jean-Christophe Léchenet |
| 2017 | Tree Buffer | VerCors | Marieke Huisman, Wytse Oortwijn |
| 2016 | Matrix Multiplication | VeriFast | Bart Jacobs |
| 2016 | Matrix Multiplication | Dafny | Luca Weibel and Christiaan Dirkx |
| 2016 | Matrix Multiplication | Why3 | Martin Clochard and Léon Gondelman and Mário Pereira |
| 2016 | Binary Tree Traversal | VeriFast | Bart Jacobs |
| 2016 | Binary Tree Traversal | Why3 | Martin Clochard and Léon Gondelman and Mário Pereira |
| 2016 | Static Tree Barrier | VeriFast | Bart Jacobs |
| 2015 | RELAXED PREFIX | Why3 | Jean-Christophe Filliâtre and Guillaume Melquiond |
| 2015 | PARALLEL GCD | Why3 | Jean-Christophe Filliâtre and Guillaume Melquiond |
| 2015 | DANCING LINKS | Why3 | Jean-Christophe Filliâtre and Guillaume Melquiond |
| 2012 | Longest Common Prefix | VeriFast | Bart Jacobs and Jan Smans |
| 2012 | Prefix-Sum | VeriFast | Bart Jacobs and Jan Smans |
| 2012 | Tree Del | VeriFast | Bart Jacobs and Jan Smans |
| 2011 | Finding the Maximum in an Array | Dafny | Julian Tschannen and Nadia Polikarpova |
| 2011 | Finding the Maximum in a Tree | Dafny | Julian Tschannen and Nadia Polikarpova |
| 2011 | Finding Two Duplets in an Array | Dafny | Julian Tschannen and Nadia Polikarpova |

## H   EXAMPLE CHALLENGE AND SOLUTION

```
1  // # Description
2  // This challenge is an instance of Kaldewaij's Search by Elimination, where an element with a
        given property is located by eliminating elements that do not have that property. The
        challenge was selected as it involves a relatively simple but interesting invariant,
        expressing that the maximal element is in the remaining search space rather than
        maintaining the maximal element found so far. A pseudo-code implementation is as follows:
3  // int max(int[] a) {
4  //     int x = 0;
5  //     int y = a.length-1;
6  //     while (x != y) {
7  //         if (a[x] <= a[y]) x++;
8  //         else y--;
9  //     }
10 //     return x;
11 // }
12 // # Task
13 // Please implement and verify that the index returned by the method max() given points to an
        element maximal in the array
14 pub fn max(a: &[int]) -> (result: usize)
15     requires
16         a.len() > 0,
17     ensures ({
18         result < a.len(),
19         forall|i: int| 0 <= i && i < a.len() ==> a[result as int] >= a[i],
20     }),
21 {
22     let mut x: usize = 0;
23     let mut y: usize = a.len() - 1;
24     ...
25     while x != y
26         invariant
27             ...
28             x as int <= max_idx && max_idx <= y as int,
29             forall|i: int| 0 <= i && i < a.len() ==> a[max_idx] >= a[i],
30             decreases y - x
31     {
32         if a[x] <= a[y] {
33             proof {
35                 ...
36                 if max_idx == x as int {
37                     assert(a[x] == a[y]);
38                     max_idx = y as int;
39                 }
40                 ...
41                 assert(x as int + 1 <= max_idx && max_idx <= y as int);
42             }
43             x += 1;
44         } else {
45             proof {
46                 ...
47             }
48             y -= 1;
49         }
50     }
51     ...
52     return x;
53 }
```

Figure 7: An example challenge stated in natural language highlighted in green and its potential solution in Verus with code implementation in grey, spec in yellow and proof in orange and invariants (a special kind of proof) in pink. This challenge is from 2011 and the solution is generated by Claude-3.7-Sonnet.

## I   DECLARATION OF LLM USAGE

This research evaluates LLM's performance on formal verification tasks. As for the paper preparation, LLM is ONLY used to polish the writing.

