# OpenReview forum: "VerifyThisBench: Generating Code, Specifications, and Proofs All at Once"
_ICLR.cc/2026/Conference — Submitted to ICLR 2026_

### Official Review · Reviewer_grL3 · 2025-10-28

**Soundness:** 3
**Presentation:** 3
**Contribution:** 2
**Rating:** 6
**Confidence:** 4

**Summary:**

This paper introduces VerifyThisBench, a challenging benchmark for evaluating LLMs' ability to perform end-to-end program verification from natural language descriptions. The benchmark requires models to simultaneously generate formal specifications, implementations in verification-aware languages, and machine-checkable correctness proofs.

**Strengths:**

- A good benchmark evaluating end-to-end verification from natural language across multiple frameworks
- Competition-grade tasks (up to 648 lines) are more challenging than prior work (Verina: 225 lines max)
- The benchmark spans seven verification tools (Dafny, Why3, VeriFast, VerCors, Frama-C, Verus, CBMC) across multiple programming paradigms and languages, providing broad assessment of verification capabilities

**Weaknesses:**

- No soundness analysis: The paper provides no detection or analysis of whether "passing" solutions achieve verification through unsound means: inserting `assume false` or `#[verifier::external_body]`
- No benchmark contamination analysis: The benchmark uses publicly available VerifyThis competition problems and solutions (2011-2024), all accessible online before model training cutoffs.
- Backwards spec-gen task design: In spec-gen tasks, models receive implementation + complete proof but must generate specifications. This contradicts realistic verification workflows where specs come first, and critically, the provided proof leaks what the specification should be since proofs explicitly reference specification properties.

**Questions:**

- In spec-gen tasks, why do you provide the proof to models when asking them to generate specifications? Doesn't the proof leak what the spec should be? In practice, the spec should be designed first and then write the proofs.
- How many "passing" solutions use unsound verification shortcuts like `assume false`, overly broad assumptions, or unsafe functions?
Have you implemented automated detection for common unsound patterns?
- Have you analyzed whether VerifyThis competition problems appear in model training data?
- Why does o3-mini, your best model on VerifyThisBench (9.37%), perform worse on the supposedly easier VerifyThisBenchXS (8.28%)?
- What percentage of your 154 benchmark tasks have verified reference solutions? Please report verification rates for all ground-truth solutions.
- How many of the "passed" solutions have specifications that are verifiable but semantically wrong (e.g., proving a weaker property than intended)? What is the performance of LLM-as-a-judge in this task?

---

> ### Author Response · Authors · 2025-11-28
>
> We thank the reviewer for your comments and questions. Below, we respond to your concerns and questions:
>
> 1) Soundness Analysis: We agree that it is always possible for the LLMs to insert assumptions to pass verification, and we have found a few rare cases where LLMs insert assumptions into lemmas, while they do not meaningfully impact the evaluation results. We modified our code to include a scan for assumptions based on the surrounding context to assess whether the assumption is grounded and updated the tables with the new values, analyzing common unsound patterns such as #[verifier::external_body], assume(false), etc, counting assumptions as failed tasks. This pipeline does not cover CBMC and Frama-C intentionally as they can contain annotations relevant to intended modelling (explained below). In total, we have found 7 unsound cases and updated the tables with these results.
>
> It is, in some cases, meaningful to allow assumptions by the LLMs, when the question requires us to do so. We investigated a subset of the generated logs and found that most LLMs in CBMC tasks assumed malloc calls returned successfully, which we believe is a fair assumption. Other cases include assuming wellformedness of data structures in certain cases, as well as question-specific assumptions. Because of these reasons, it is non-trivial to simply forbid assumptions for certain tools. However, even with such assumptions (where appropriate), the models fail to achieve meaningful results.
>
> 2) Contamination Analysis: While contamination is possible and we treat it as a limitation, we believe it is unlikely to explain our overall results for two reasons: First, solutions for certain parts of our dataset were developed manually recently over the past few months, and were only recently made public. Particularly, for Verus, we have developed our own solutions for the past VerifyThis challenges. Second, the models fail to achieve a meaningful accuracy, showing that contamination, if any, fails to significantly skew the results. These observations suggest contamination alone is unlikely to account for the low success rates and common failure modes we observe.
>
> 3) Regarding Backwards spec-gen task design: We agree that providing a proof leaks aspects of the intended specification, which is a part of the reason why VerifyThisBenchXS is considered a relaxation. The full VerifyThisBench benchmark requires implementing the entire specification, code, and proofs, all at once without revealing any details about any part. Specification-Gen task only exists as a relaxation for the benchmark in VerifyThisBenchXS, since we have shown that the models fail to achieve meaningful accuracy, and is unrealistic by design to help break down the benchmark into easier subtasks. We maintain the same relaxation principle in Loop-Gen and Code-Gen tasks. Our main benchmark is VerifyThisBench, which provides no relaxations/hints, and is  independent from VerifyThisBenchXS which provides those relaxations.
>
> 4) Discrepancy in o3-mini: We believe the difference is small in absolute terms and is within the expected variance. Due to cost constraints, we were only able to run our benchmark once on all models, and evaluation results may change slightly as a result of running the benchmark multiple times.
>
> 5) Regarding solutions: All questions included in our benchmark have ground truth solutions, which are verified by the respective verifiers. These are taken from VerifyThis solutions, and include the solutions we have developed (in Verus).
>
> 6) LLM-as-a-Judge performance: We investigate a subset of the results from using LLM-as-a-Judge (in addition to formal verification) manually, and find that most models except for o3-mini are accurate in their self-assessment. In the examined subset, we did not observe any false negatives. Additionally, for Specification-Gen task in VerifyThisBenchXS, the performance of LLM-as-a-Judge is less relevant as the specification is inserted into the code, and providing an incorrect specification is often caught by the verifier as the proof obligations fail.
>
> We believe verifying a formal specification is correct from natural language is a non-trivial task and there exists no tool that can automatically achieve this. In real life workflows, this is done via humans (programmers) as judges, who decide whether the definitions or the specifications are correct. For this reason, we believe using LLM-as-a-Judge (in addition to formal verification) is appropriate for the task, and add that it is difficult to evaluate the performance of the judges quantitatively apart from manually inspecting a subset.

---

### Official Review · Reviewer_iGij · 2025-11-01

**Soundness:** 1
**Presentation:** 3
**Contribution:** 2
**Rating:** 2
**Confidence:** 4

**Summary:**

This paper presents a benchmark set for generating fully verified code from a natural language prompt. The benchmarks are based on tasks drawn from the long-running VerifyThis Competition Series. The paper evaluates a range of existing models on the benchmarks for a wide range of verification frameworks and shows that the benchmarks are very challenging.

**Strengths:**

- Given the increasing usage of LLM in writing code, evaluating their ability to write verifiably correct code is a timely topic.
- I find it a very nice and clever idea to use the tasks described in VerifyThis competition. This choice helps to focus on tasks that existing verification tools are more likely to be able to handle.
- The evaluation considers a wide range of existing models and is therefore quite thorough.

**Weaknesses:**

- My main issue with the benchmark is that the success rate is not a very meaningful metric. If I understand correctly, the LLM can achieve 100% success rate by generating a trivial and fully verified program that is completely irrelevant to the natural language prompt. Certainly, LLMs are trained to follow instructions. However, the evaluation metric is too easy to game. Imagine having a leaderboard based on success rate for VerifyThisBench, it would be really unclear whether top performers are simply generating easily verifiable programs that do not (fully) follow the natural language instructions, or whether they are actually performing well.
- The author did try to mitigate the issue above by proposing a coherence check based on LLM self-assessing its generated artifact. Self-assessed coherence is something that can be easily gamed; the model is incentivized to just lie. So in short, one can easily get 100% success rate and 100% self-assessed coherence by generating a trivial and verified program and lying about the coherence. As a developer of practical LLMs, this is not good practice. However, a benchmarking framework should be designed to have a level of robustness against these gaming strategies.
- There are different potential ways to mitigate the problems above. 1) Instead of self-assessment, perhaps assess using an independent LLM or even employ mutual assessment; 2) one could associate each benchmark with a set of test cases; 3) This would be much harder, but ideally, the evaluation could perform some equivalence checking between the generated code and the ground truth, or the generated specification and the ground specification. 1) and 2) are both not ideal. 2) is probably better than 1). I acknowledge that the paper did explore 2) to a limited degree, but it does not apply it to all benchmarks. If something along the line of 3) can be achieved, then I believe the benchmark set would differentiate itself significantly from concurrent work and consitute a solid evaluation framework.

**Questions:**

N/A

---

> ### Author Response · Authors · 2025-11-28
>
> Thank you for your thoughtful comments and suggestions to improve our evaluations. Below we respond to each one of your concerns:
>
> 1) Regarding LLMs gaming the benchmark: We agree that in an unconstrained setting, the LLMs can game the full benchmark. As a mitigation, we include the full benchmark (VerifyThisBench) along with a relaxation, VerifyThisBenchXS. Because the full benchmark is unconstrained, misaligned solutions are possible in principle, but VerifyThisBenchXS mitigates this by fixing reference artifacts (proofs and/or code), thus constraining the space of possible output specifications. We therefore use (i) coherence checking and (ii) VerifyThisBenchXS to provide additional evidence about alignment and to isolate component difficulty. Specification-Gen, as a relaxation, provides the proof and the full code, and expects the LLM to generate the specification, thus measuring the model's ability to generate specifications. In Specification-Gen, the model must produce a specification that is consistent with the provided reference implementation and proofs. Specifications that do not align with these reference artifacts typically fail verification due to unmet proof obligations and implicit conditions (e.g., index bounds, preconditions of library functions).
>
> A similar relationship exists for Code-gen and Loop-gen. If the LLM does not write code that accurately reflects the specification, the existing proofs and specifications would fail, which would be caught by the verifier. To summarize, VerifyThisBenchXS reduces opportunities for misaligned specifications by constraining generation and enforcing consistency against fixed reference artifacts. In the full VerifyThisBenchmark, the results can always be cross-compared with VerifyThisBenchXS tasks, which would help diagnose which LLMs produce specifications inconsistent with these programs and/or proofs.
>
> 2) Regarding the coherence check: apart from the mitigations we provide as explained in 1) above, we believe coherence check is a good option to automatically evaluate the generated specifications. However, we agree with the reviewer that testing the specifications would provide further mitigations, though this would be impractical to integrate (due to the lack of testing in many software verification tools we use). Additionally, natural language specifications can be ambiguous given constraints for languages (such as, whether the model should consider integer over/underflows in Verus, when to return Option type, etc.) which raises the question of how testing would consider such cases.
>
> In general, we believe LLM-as-a-Judge performance (in addition to formal verification) can be a good choice in this context for multiple reasons: 1) Natural language is ambiguous and there is no one-fits-all testing method for specifications in verification languages, 2) in real life applications of formal methods, this ambiguity between natural language and formal specification is usually resolved by assuming a human checks the formal definition (human-as-a-judge), an idea akin to our LLM-as-a-judge, 3) the specifications / coherence can always be cross-compared with results from VerifyThisBenchXS as explained above.
>
> 3) Finally, regardless of these concerns, LLMs still get a very low accuracy on VerifyThisBench and VerifyThisBenchXS.
>
> We thank you again for your valuable suggestions and feedback.

---

### Official Review · Reviewer_5XC4 · 2025-11-04

**Soundness:** 3
**Presentation:** 3
**Contribution:** 3
**Rating:** 6
**Confidence:** 4

**Summary:**

This paper introduces VerifyThisBench, benchmark for evaluating LLM coders on end-to-end formal program verification. The benchmark is derived from the VerifyThis competition series, and includes generating code, formal specifications, and machine-checkable proofs simultaneously from natural language descriptions. The authors evaluate nine models (including both closed frontier models and leading open models) across seven verification frameworks (Dafny, Verus, Frama-C, etc.) on over 700 tasks. The results show that performance is quite low, although iterative improvement seems to offer potential for significant gains.  To isolate sources of difficulty, the authors introduce VerifyThisBenchXS, a relaxed variant providing partial implementations or proofs, where performance improves moderately but still demonstrates substantial gaps.

**Strengths:**

The paper's core strength is its novel and rigorous benchmark, VerifyThisBench, which addresses a critical gap in LLM evaluation by moving beyond simple functional testing.

-- Unlike most benchmarks, it requires models to generate specifications, code, and machine-checkable proofs from only a natural language prompt.

-- It introduces a unified environment for seven distinct verification tools (e.g., Dafny, Verus, Frama-C), a significant step up from single-framework benchmarks like Verina and Clever.

-- The sourcing is from the VerifyThis competition, which is arguably more "real-world" than previous benchmarks in this space.

-- I appreciated the design of VerifyThisBenchXS, and the analysis in isolating specific failure modes.

**Weaknesses:**

The paper's core premise is to evaluate verification from natural language, but doesn't really directly measure this goal.  The main quantitative experiments check if the generated proof is valid for the generated specification.  The closest result in the main paper is in Section 4.5, where "coherence" is measured (whether the LLM believes the generated spec matches the natural language description).  Thus, I believe the abstract & intro need to be somewhat repositioned, or is otherwise a bit misleading in this respect.

The extremely low success rates on this benchmark can also need to noise, where "randomly" getting a few extra problems correct can lead to huge relative gains in improvement.  Worth doing a more detailed analysis on variance/reproducibility of results.  The common approach is to run the same setup several times and observe the variance.  I realize this is expensive, but could be worth doing for a few LLMs.

**Questions:**

No further questions, other than commenting on my weaknesses.

---

> ### Author Response · Authors · 2025-11-28
>
> We thank the reviewer for your thoughtful comments. Below is our response and actions:
>
> 1) Regarding evaluating verification from natural language: The core goal of our paper is to evaluate the end-to-end program verification from natural descriptions. We agree that our main metric (verifier pass rate) does not, by itself, fully measure semantic alignment with the natural language prompt, since it primarily checks proof validity for the generated specification. We have revised the abstract to emphasize this point, and will revise the introduction to distinguish these cases.
>
> While coherence in section 4.5 is included to evaluate the generated specification, we further evaluate code and proof generation, and show that LLMs struggle to achieve a reasonable accuracy for all of those tasks together, thus showing that models struggle to produce a verifier-accepted program (including specifications, code, and proofs) end-to-end. The tasks do not end only at specification generation, but include further generation for code and proofs.
>
> Additionally, automatically verifying the semantic alignment between generated specifications and baseline specifications is technically challenging to do reliably at scale, as there are multiple different ways to encode the same natural language specification. This is why we introduce a coherence metric, as there is no reliable way to assess the scope and intent of the natural language (in real life applications of formal methods, this is judged by humans), and formal specifications, nor a general, tool-independent way to verify equivalence between two formal specifications. For this reason, we introduce the coherence metric and use LLM-as-a-Judge, to appropriately reflect the human behaviour.
>
> 2) Regarding your comment on randomness: We agree that such cases may occur, but should not meaningfully impact the results presented in our table. We have repeated the full VerifyThisBench, zero-shot experiment 3 different times for gpt-4o-mini and Verus, and gotten the following accuracies: 7.14%, 4.55%, 5.84%. The original value presented in the table is 6.49%. We acknowledge that repeating the experiment multiple times may change the results, but we believe overall results will be similar, not affecting our conclusions. Due to cost constraints, we are not able to repeat the full experiment across all tasks and models.

---

### Meta-Review · Area_Chair_1FQ8 · 2026-01-05

**Summary:**

The paper proposes a new benchmark for evaluating end-to-end program generation, including code, formal specifications, and machine-checkable proofs generated from natural language descriptions. All reviewers agree that the benchmark is timely, more challenging due to its use of VerifyThis problems, and more scalable by supporting multiple verification tools.

The main concern is that the primary success-rate metric does not adequately reflect the quality of specification generation. Reviewer 1 notes that success only guarantees that the proof is valid for the generated specification, not that the specification matches the task. Reviewer 2 points out that a model could generate a trivial or irrelevant program that still passes the verifier. Reviewer 3 suggests that assumption statements in proofs can also pass the verifier without completing the proof. All reviewers agree that self-assessed coherence is problematic for evaluating specification quality.

In the rebuttal, the authors argue that self-assessment is reasonable based on manual inspection of a subset of examples, showing most models are accurate. While acknowledging that automatically verifying specification correctness is difficult. However, this response is not fully convincing. As Reviewer 2 notes, a benchmark should be robust against such gaming strategies, and the current evaluation does not sufficiently prevent models from cheating. As a result, the rebuttal does not resolve the core concerns.

Given two borderline ratings and one negative rating, I lean toward rejection.

**Reviewer Concerns:**

Reviewer 1’s concern about the large variance in the results is partially addressed by the additional three runs of GPT-4o-mini provided in the rebuttal. Reviewer 3’s concern regarding the design of the backward specification-generation task may also be partially addressed.

The core concerns about the evaluation of specification generation remain unaddressed. Reviewer 3’s concern about data contamination is also not addressed.

**Reviewer Scores:**

Given that the major issues are not easily resolvable, reviewers are likely to keep their scores unchanged.

---

### Decision · Program_Chairs · 2026-01-26

Reject